# Genome-Wide Identification and Expression Analysis of the *High-Mobility Group B* (*HMGB*) Gene Family in Plant Response to Abiotic Stress in Tomato

**DOI:** 10.3390/ijms25115850

**Published:** 2024-05-28

**Authors:** Jinhui Zheng, Huimeng Tang, Jianquan Wang, Yue Liu, Lianjing Ge, Guobiao Liu, Qinghua Shi, Yan Zhang

**Affiliations:** College of Horticultural Science and Engineering, Shandong Agricultural University, Tai’an 271018, China; 2022110324@sdau.edu.cn (J.Z.); 2020010081@sdau.edu.cn (H.T.); dream19980613@163.com (J.W.); 2021010059@sdau.edu.cn (Y.L.); 2022010067@sdau.edu.cn (L.G.); 15610306032@163.com (G.L.); qhshi@sdau.edu.cn (Q.S.)

**Keywords:** genome-wide identification, expression pattern, high-mobility group B (HMGB), plant growth, abiotic stress, tomato

## Abstract

High-mobility group B (HMGB) proteins are a class of non-histone proteins associated with eukaryotic chromatin and are known to regulate a variety of biological processes in plants. However, the functions of *HMGB* genes in tomato (*Solanum lycopersicum*) remain largely unexplored. Here, we identified 11 members of the *HMGB* family in tomato using BLAST. We employed genome-wide identification, gene structure analysis, domain conservation analysis, *cis*-acting element analysis, collinearity analysis, and qRT-PCR-based expression analysis to study these 11 genes. These genes were categorized into four groups based on their unique protein domain structures. Despite their structural diversity, all members contain the HMG-box domain, a characteristic feature of the HMG superfamily. Syntenic analysis suggested that tomato SlHMGBs have close evolutionary relationships with their homologs in other dicots. The promoter regions of *SlHMGBs* are enriched with numerous *cis*-elements related to plant growth and development, phytohormone responsiveness, and stress responsiveness. Furthermore, *SlHMGB* members exhibited distinct tissue-specific expression profiles, suggesting their potential roles in regulating various aspects of plant growth and development. Most *SlHMGB* genes respond to a variety of abiotic stresses, including salt, drought, heat, and cold. For instance, *SlHMGB2* and *SlHMGB4* showed positive responses to salt, drought, and cold stresses. *SlHMGB1*, *SlHMGB3*, and *SlHMGB8* were involved in responses to two types of stress: *SlHMGB1* responded to drought and heat, while *SlHMGB3* and *SlHMGB8* responded to salt and heat. *SlHMGB6* and *SlHMGB11* were solely regulated by drought and heat stress, respectively. Under various treatment conditions, the number of up-regulated genes significantly outnumbered the down-regulated genes, implying that the *SlHMGB* family may play a crucial role in mitigating abiotic stress in tomato. These findings lay a foundation for further dissecting the precise roles of *SlHMGB* genes.

## 1. Introduction

High-mobility group (HMG) proteins were discovered in mammalian cells and named for their mobility in polyacrylamide gel electrophoresis [1]. Bustin et al. (2001) subdivided them into three superfamilies: HMGA, HMGB, and HMGN. The members containing the AT-hook domain belong to the HMGA family; those containing the HMG-box domain are assigned to the HMGB family; and those containing nucleosomal binding domain are categorized as the HMGN family [2].

In plants, HMGB proteins widely exist, and their HMG-box domain mainly consists of three α-helices arranged in an L-shape and is flanked by alkaline N-terminal and highly acidic C-terminal domains, which regulate DNA bending or binding [3,4,5]. The HMG-box can be found in HMGB, STRUCTURE-SPECIFIC RECOGNITION PROTEIN 1 (SSRP1), and AT-RICH INTERACTIVE DOMAIN (ARID) proteins, which appear to be plant specific [6,7].

At present, the *HMGB* genes have been cloned from a variety of plant species, including *Arabidopsis* [8,9], maize (*Zea mays*) [10,11], soybean (*Glycine max*) [12], wheat (*Triticum aestivum*) [13], pea (*Pisum sativum*) [14], and rice (*Oryza sativa*) [15,16]. The biological functions of some of these genes have been characterized. For instance, *Arabidopsis AtHMGB15* controls pollen tube growth by interacting with the transcription factors responsible for pollen and pollen tube development. The *athmgb15-1* mutant exhibits abnormal pollen tubes and reduced fertility [17]. *AtHMGB8* and *AtHMGB9* regulate cell division, such as chromosome condensation and/or segregation [7]. In mammals, extracellular HMGB1, a prototypical damage-associated molecular pattern (DAMP) molecule, functions in activating the immune response to prevent infection and promote healing after tissue injury. Interestingly, HMGB1 not only stimulates tissue repair and regeneration but also induces inflammatory response and triggers cellular apoptosis [18]. In plants, the extracellular HMGB3 of *Arabidopsis* exhibits a similar DAMP function. When the plants are infected by pathogens, HMGB3 is released into the extracellular space, triggering a series of typical innate immune responses. This reveals a novel function of plant HMGB proteins [19].

Furthermore, *HMGB* can regulate plant growth and development under stress conditions. The ectopic expression of maize *HMGB1* affects primary root growth in tobacco seedlings due to the accumulation of small cells in the cytokinetic zone of the roots of transgenic plants, resulting in a reduction in primary root length. However, this phenotype is transient and gradually disappears with plant senescence [20]. In *Arabidopsis*, either the mutation or overexpression of *AtHMGB1* results in a reduction in primary roots, while the aboveground plant remains unaffected [21]. Under salt and drought stress, *Arabidopsis* plants overexpressing *AtHMGB2* or lacking *AtHMGB5* show delayed germination and subsequent growth, but the overexpression of *AtHMGB4* does not affect seed germination and plant growth [22]. Under high temperature and drought stress, ectopic expression of cucumber *CsHMGB* delays seed germination in *Arabidopsis* but does not affect postgermination growth [23]. Cold stress leads to a significant increase in the transcript levels of *AtHMGB2*, *AtHMGB3*, and *AtHMGB4*, and drought or salt stress down-regulates the expressions of *AtHMGB2* and *AtHMGB3*; however, the molecular mechanism of this process is not clear [23].

In addition, OsHMGB1 in rice can interact with SWEET (Sugars Will Eventually be Exported Transporter) proteins to negatively regulate plant resistance to rice leaf blight pathogens. Meanwhile, it may also take part in the negative regulation of the immune response by interacting with the calmodulin-like protein OsCML3 to regulate the Ca^2+^ signaling pathway [24]. OsHMGB707 positively regulates drought resistance in rice by directly binding to the promoter of *OsDREB1G*, and the expression of stress-related genes is increased in *OsHMGB707* overexpressing lines [25]. In cucumber, the silencing of *CsHMGB* reduces plant resistance to propamocarb, as evidenced by the increased malondialdehyde (MDA) content and reactive oxygen species (ROS) content, accompanied by decreased activity of antioxidant enzymes and the ascorbate–glutathione (AsA-GSH) system. In contrast, *CsHMGB* overexpression promotes GSH-dependent detoxification, improves antioxidant capacity, and reduces ROS accumulation [26]. The induction of *BnHMGB2* isolated from oilseed rape in response to low-temperature stress potentially plays a key role in plant resistance to low-temperature stress. The 11 *HMGB* genes identified in *Betula platyphylla* respond to salt, osmotic stress, and abscisic acid (ABA) treatments, and the overexpression of *BpHMG6* increases antioxidant enzyme activity along with enhanced ROS scavenging under salt stress, leading to reduced plant damage and death caused by salt stress [27].

Taken together, the *HMGB* genes have been studied in some plant species. However, to date, no reports have demonstrated the functions of any *HMGBs* in tomato. Therefore, we conducted this study and identified the *HMGB* members in tomato. We then analyzed the gene structures, conserved domains, phylogenetic relationships, *cis*-elements, expression profiles in different tissues and abiotic stress conditions, and subcellular localization in detail. These data suggest the potential roles of tomato *HMGB* genes in plant growth and response to abiotic stresses.

## 2. Results

### 2.1. Identification and Chromosome Localization of the HMGB Genes in Tomato

By BLAST search and conserved domain analysis, a total of 11 tomato *HMGB* genes were identified and subsequently named according to their location on the chromosome. According to the GFF genome annotation file, the genomic sequence length of *SlHMGBs* is 1556–14,596 bp, the coding sequence (CDS) length is 420–2007 bp, the encoded amino acid length is 139–668 aa, and the molecular weight is 15.52–74.46 kDa. SlHMGB3, SlHMGB4, SlHMGB6, SlHMGB8, and SlHMGB9 are acidic, and the others are basic. The subcellular prediction according to WoLF PSORT showed that all SlHMGB proteins are potentially localized in the nucleus (Table 1).

In addition, we performed chromosomal localization of *SlHMGBs,* which showed that they are distributed on seven chromosomes of tomato. Precisely, *SlHMGB1*, *SlHMGB6*, and *SlHMGB7* are localized on chromosome 1, chromosome 4, and chromosome 8, respectively. Additionally, two genes mentioned in the parenthesis are localized on each of chromosome 2 (*SlHMGB2* and *SlHMGB3*), chromosome 3 (*SlHMGB4* and *SlHMGB5*), chromosome 9 (*SlHMGB8* and *SlHMGB9*), and chromosome 12 (*SlHMGB10* and *SlHMGB11*) (Figure 1).

To gain a deeper understanding of the evolutionary clues about SlHMGB members, we conducted covariance analysis between tomato and other plant species, including five dicotyledonous and one monocotyledonous plant. Many genes were identified in these species that showed covariance with the tomato *HMGB* genes, such as eleven pairs in tomato and *Arabidopsis*, ten pairs in tomato and *Solanum pennellii*, nine pairs in tomato and tobacco (*Nicotiana tabacum*), ten pairs in tomato and pepper (*Capsicum annuum*), eleven pairs in tomato and potato (*Solanum tuberosum*), and three pairs in tomato and rice (*Oryza sativa*) (Figure 2, Appendix A). After removing duplicated genes, we found that nine, eight, eight, six, and six *SlHMGBs* are homologous to *HMGB* genes of potato, *Solanum pennellii*, pepper, *Arabidopsis*, and tobacco, respectively, whereas only three *SlHMGB* members are homologous to rice *HMGB* genes, suggesting a closer evolutionary relationship of *HMGBs* between tomato and dicotyledonous plants. Only two members, *SlHMGB2* and *SlHMGB7*, showed covariation of genes in all five dicotyledonous plants and rice (Appendix A). In addition, *SlHMGB2* and *SlHMGB7* and their covariance genes might have evolved from the coevolution of dicotyledons and monocotyledons, whereas other *SlHMGB* members might have a common origin in dicotyledons.

### 2.2. Gene Structure and Conserved Domains of SlHMGBs

The structures of the *SlHMGB* genes were obtained by comparing the CDS sequence with the genomic sequence using the GSDS online website. The analysis of gene structures shows that the number of exons in the *SlHMGB* gene family ranges from 5 to 14, with most of the genes containing seven exons and two genes (*SlHMGB3* and *SlHMGB5*), possessing more than ten exon numbers, and *SlHMGB11* contains only five exons (Figure 3).

In addition, protein domain analysis of SlHMGBs and multiple sequence alignment of tomato, *Arabidopsis*, and rice HMGB proteins were performed. Based on the results, we categorized the SlHMGB family into four groups. Group I includes five members, SlHMGB2, 4, 5, 6, and 8, which have only one HMG-box domain; group II has one member, SlHMGB7, that contains three HMG domains; SlHMGB1, 9, 10, and 11, possessing one ARID domain and one HMG-box, are assigned to group III; and SlHMGB3 belonging to group IV has one SSRP1 domain and one HMG-box (Figure 4A and Appendix A). The conserved motifs of SlHMGBs were further investigated by the MEME website (Version 5.5.5), and the sequence logos showed that the ARID-domain and HMG-box exhibit low conservation levels among SlHMGB proteins (Figure 4B).

### 2.3. Phylogenetic Analysis of HMGB Members in Tomato

To investigate the evolutionary relationship among HMGB homologs, a total of eighty-five HMGB proteins from eight plant species, including *Solanum lycopersicum* (Sl), *Arabidopsis thaliana* (At), *Oryza sativa* (Os), *Solanum melongena* (Sm), *Capsicum annuum* (Ca), *Solanum pennellii* (Sp), *Nicotiana tabacum* (Nt), and *Solanum tuberosum* (St), were obtained and used for phylogenetic tree construction. As shown in Figure 5, these HMGB homologs were divided into four groups. In the tomato SlHMGB family, there is one protein in each of group II and group IV, four members in group III, and five proteins in group I, which are consistent with the results of protein domain analysis and sequence alignment (Figure 4A and Appendix A).

### 2.4. Cis-Acting Element Analysis of SlHMGB Promoters

We extracted the 2 kb promoter sequences of *SlHMGB* genes using Tbtools (Version 2.096) and submitted them to PlantCARE (Version 1) for the identification of *cis*-acting elements. In addition to the common promoter elements, we obtained a total of twenty different types of *cis*-acting elements, which can be categorized into three major groups: plant growth and development, phytohormone responsiveness, and stress responsiveness. The CAT-box, circadian, O2-site, CCGTCC-box, GCN4_motif, and RY-element, which are involved in meristem expression, circadian control, zein metabolism, meristem activation, endosperm expression, and seed development, respectively, belong to the plant growth and development category. The phytohormone responsiveness group includes the elements of ABRE responsible for abscisic acid response, TGACG-motif and CGTCA-motif for methyl jasmonate response, P-box for gibberellin response, TCA-element for salicylic acid response, and AuxRR-core and TGA-element for auxin response. In the stress responsiveness category, the elements of ARE important for anaerobic induction response, STRE for stress response, TC-rich repeats for defense response, WUN-motif for wound response, MBS and DRE core for dehydration response, and LTR for low-temperature response are detected. Additionally, each *SlHMGB* gene contains different types and numbers of *cis*-acting elements; however, the elements related to phytohormone responsiveness or stress responsiveness are more than those related to plant growth and development in most *SlHMGB* genes (Figure 6).

### 2.5. Analysis of Tissue-Specific Expression Patterns of SlHMGB Genes

To predict the potential functions of the *SlHMGB* genes, we performed qRT-PCR analysis on different tomato tissues, including roots, stems, leaves, flowers, and fruits at different developmental stages. Then, a heatmap was constructed based on the results of qRT-PCR using TBtools software (Version 2.096). As shown in Figure 7, both *SlHMGB1* and *SlHMGB11* had the highest expression in the fruits at the mature green stage (MG-Fr). *SlHMGB6*, *SlHMGB7*, and *SlHMGB9* were mainly expressed in the leaves, whereas *SlHMGB6* and *SlHMGB7* also showed higher expressions in the stems and immature fruits (IM-Fr), and *SlHMGB9* had a higher transcript level in the flowers. *SlHMGB2*, *SlHMGB4*, and *SlHMGB8* displayed the highest expression in the fruits at the turning stage (T-Fr) and the lowest expression in the roots and stems. *SlHMGB3* and *SlHMGB5* were principally distributed in the leaves, flowers, and fruits at the immature and mature green stages. Moreover, a higher expression of *SlHMGB10* was detected in the flowers and fruits at all stages of maturity. Taken together, *SlHMGB* members show diverse tissue-specific expression profiles, indicating their potential roles in different aspects of plant growth and development, and there may be functional redundancy and divergence among these genes.

### 2.6. Expression Analysis of SlHMGB Genes under Abiotic Stress

The *cis*-acting elements of *SlHMGB* promoters were detected to be related to stress responsiveness, suggesting that *SlHMGB* genes may play important roles in plant response to environmental stress (Figure 6). Therefore, we further analyzed the expressions of *SlHMGB* genes under salt, drought, heat, and cold treatments. Overall, the *SlHMGB* members showed different expression patterns under different abiotic stresses. For example, under salt treatment, six genes, *SlHMGB2*, *SlHMGB3*, *SlHMGB4*, *SlHMGB5*, *SlHMGB8*, and *SlHMGB10,* were up-regulated, and all the peaks occurred at 9 h after treatment (Figure 8A). In addition, the other five genes showed disordered expression patterns after salt treatment (Appendix A). Upon imposition of drought stress by PEG-6000, six *SlHMGB* genes showed up-regulation in their expression patterns. With a few exceptions, the expression levels of *SlHMGB1*, *SlHMGB2*, *SlHMGB4*, *SlHMGB5*, *SlHMGB10*, and *SlHMGB11* were higher than the control during the whole treatment period, but their peaks occurred at different time points (Figure 8B). Moreover, the additional five *SlHMGB* members displayed inconsistent expression trends at various time points under PEG treatment (Appendix A).

Furthermore, six *SlHMGB* genes responded to heat stress (42 °C); among them, five members were up-regulated and one member was down-regulated. In detail, the increased expressions of *SlHMGB1*, *SlHMGB3*, *SlHMGB5*, and *SlHMGB10* were discovered at each time point after heat treatment, while *SlHMGB8* expression was induced at most time points but was unchanged at 12 h. The transcript level of *SlHMGB6* was reduced during the whole treatment period (Figure 9A). The remaining five *SlHMGB* members showed confused expression profiles under drought stress (Appendix A). Under cold stress (4 °C), three and one *SlHMGB* genes were positively and negatively regulated, respectively. The expressions of *SlHMGB2*, *SlHMGB4*, and *SlHMGB10* were increased at partial time points and unchanged at other points compared with the control, while the *SlHMGB5* level was decreased throughout the treatment period (Figure 9B). The rest of the *SlHMGB* members exhibited disorganized expression patterns (Appendix A).

Based on the *SlHMGB* members involved in plant response to the four abiotic stresses, we produced Venn plots (Appendix A). Two genes out of eleven *SlHMGB* family members, *SlHMGB5* and *SlHMGB10*, responded to all four abiotic stresses, wherein *SlHMGB10* was up-regulated by different treatments, suggesting that this gene might play key roles in tomato stress tolerance. Furthermore, both *SlHMGB2* and *SlHMGB4* positively responded to salt, drought, and cold stresses. *SlHMGB1*, *SlHMGB3*, and *SlHMGB8* were positively involved in response to two stresses (*SlHMGB1* for drought and heat stresses, *SlHMGB3* and *SlHMGB8* for salt and heat stresses). *SlHMGB6* and *SlHMGB11* were only regulated by drought or heat stress, respectively. In conclusion, most *SlHMGB* genes probably played important roles in mitigating abiotic stresses in tomato, despite showing different roles under different stresses.

### 2.7. Subcellular Localization of SlHMGB Proteins

According to the subcellular localization prediction results, all SlHMGB proteins are potentially localized in the nucleus (Table 1). To verify these results, we selected three SlHMGB proteins, SlHMGB2, SlHMGB3, and SlHMGB4, for subcellular localization in tobacco leaves. As shown in Figure 10, SlHMGB2-GFP, SlHMGB3-GFP, and SlHMGB4-GFP fusion proteins were merged with DAPI (a nucleus marker), respectively, indicating that all three proteins were localized to the nucleus, which is consistent with the predicted results. As the control, the 35S:GFP signal was distributed throughout the cell.

## 3. Discussion

Until now, the *HMGB* genes have been extensively identified in many plant species, such as *Arabidopsis* [8,9], rice [15,16], maize [10], *Vicia faba* [28], *Canavalia gladiata* [29], *Pharbitis nil* [30], soybean [12], and pea [14]. However, *SlHMGBs* in tomato have received little attention to date. In this study, we identified a total of 11 tomato *HMGB* genes (Table 1), which were named *SlHMGB1-SlHMGB11* based on their locations on the chromosome. Through protein structural domain analysis and amino acid multiple sequence alignment, we found that SlHMGB proteins possess a conserved HMG-box domain (Figure 4), a typical structure of the HMGB family. This domain can mediate non-sequence-specific or sequence-specific DNA binding [6]. However, multiple sequence alignment and motif analyses revealed that the HMG-box domains exhibit low similarity among HMGB homologs of different plant species, as well as among different SlHMGB members in tomato (Appendix A and Figure 4B). This suggests that HMGBs display low conservation during evolution.

Furthermore, the phylogenetic tree categorizes the HMGB members into four groups based on the structural domains. Group I includes the members containing the HMG-box, group II has the members containing the 3xHMG-box, group III possesses the members containing the ARID domain, and group IV has the members containing the SSRP1 box (Figure 5). The *HMGB* gene pairings were compared between tomato and other species, such as *Arabidopsis*, rice, tobacco, *Solanum pennellii*, pepper, and potato (Figure 2, Appendix A). The results showed that *SlHMGB* exhibits a close evolutionary relationship with its homologs in dicotyledonous plants. In addition, *SlHMGB2* and *SlHMGB7* have syntenic pairs in both dicotyledons and monocotyledons (Appendix A), suggesting that these homologs may have existed prior to divergence.

*HMGB* genes have previously been proved to modulate plant growth and development, such as seed germination [23], leaf growth, flowering induction [31], pollen maturation, pollen tube development [17], reduction in seed dormancy [32], and anthocyanin accumulation [33]. Herein, our results showed that the *SlHMGB* genes exhibit diverse expression patterns in different tomato tissues (Figure 7). Expressions of *SlHMGB* genes are generally low in roots, except *SlHMGB9*. Two members are highly expressed in the stems, eight members in the leaves, nine members in the flowers, five members in the IM-fruits, five members in the MG-fruits, four members in the T-fruits, and two members in the MR-fruits. This implied the potentially important roles of the *SlHMGB* family in the growth and development of different organs. Moreover, some *SlHMGB* members show similar expression patterns, indicating they may have redundant functions in plant growth and development of tomato.

Moreover, *HMGB* genes have been functionally characterized in some plant species, especially *Arabidopsis* [13,14,15,16,29,34,35,36]. Here, we deduced the functional similarity and discrepancy between tomato and *Arabidopsis HMGB* homologs based on the tissue-specific expression patterns (Figure 7), syntenic analyses (Figure 2, Appendix A), and previous studies. For example, *AtHMGB2* is syntenic with *SlHMGB2* and *SlHMGB4* (Figure 2), and *SlHMGB2* and *SlHMGB4* are up-regulated under salt, drought, and cold stresses (Figure 8 and Figure 9), suggesting that they may play important roles in mitigating abiotic stress in tomato, whereas *AtHMGB2* expression is induced by cold stress and inhibited by drought and salt stresses [23], indicating that *AtHMGB2* also responds to abiotic stress, but the effects are different from *SlHMGB2* and *SlHMGB4* under diverse stresses. *SlHMGB9* is principally expressed in the flowers (Figure 7), revealing its potential involvement in flower development. Correspondingly, its syntenic gene, *AtHMGB15,* has been demonstrated to play crucial roles in pollen development and pollen tube growth in *Arabidopsis* [17]. This suggests that there may be similar functions between *SlHMGB9* and *AtHMGB15*. Based on these findings, we speculated that tomato *SlHMGB* genes may exhibit conserved and distinct functions with their homologs of other plant species.

In *Arabidopsis*, cold stress results in a significant increase in the transcript levels of *AtHMGB2*, *AtHMGB3*, and *AtHMGB4*, while drought or salt stress suppresses the expression of *AtHMGB2* and *AtHMGB3* [23]. By analyzing the qPCR results in this study, we found that tomato *SlHMGB* genes also respond to abiotic stresses, for example, salt treatment causes differential expressions of six genes, all of which are up-regulated (Figure 8A), while the drought and heat treatments lead to increased expressions of five genes and decreased expression of one gene (Figure 8B and Figure 9A). Three genes are positively regulated, and one gene is negatively regulated by cold treatment (Figure 9B). Overall, the number of up-regulated *SlHMGB* genes significantly outnumbers the down-regulated genes under various abiotic stresses, suggesting that the *SlHMGB* gene family may play a crucial role in mitigating abiotic stress in tomato. In addition, Venn diagrams of the qPCR results showed that two *SlHMGB* genes are involved in response to four abiotic stresses (Appendix A), of which *SlHMGB10* is significantly up-regulated under all treatments (Figure 8 and Figure 9). This indicates that *SlHMGB10* may play a critical role in enhancing stress tolerance in tomato. In conclusion, the tomato *SlHMGB* family genes are extensively involved in response to multiple abiotic stresses.

## 4. Materials and Methods

### 4.1. Plant Materials, Growth Conditions, and Treatments

The tomato cultivar Micro-Tom was used in this study. The plants were cultivated in a greenhouse at Shandong Agricultural University (Tai’an, China; 34° N, 117° E) with a daytime temperature of 25 °C for 16 h and a nighttime temperature of 18 °C for 8 h. Field management was carried out following the standard practices.

To analyze the tissue-specific expression, the roots, stems, and leaves from the plants at the four true-leaf stage, fully opened flowers, and fruits at the immature (IM, 15 days post anthesis), mature green (MG), turning (T), and mature red (MR) stages were collected for RNA extraction. Three biological replicates were prepared for each tissue.

For salt stress, the tomato seedlings were cultivated in Hoagland’s nutrient solution and then subjected to 150 mM salt treatment at the four true-leaf stage. At 0 h, 3 h, 6 h, 9 h, 12 h, and 24 h after treatment, the leaves were collected, respectively. Additionally, the plants cultivated in the pots containing the matrix were prepared for drought, heat, and cold treatments, and the matrix conditions (including volume and composition) are the same in every pot under each treatment. For drought stress, the plants at the four-week-old stage were watered with a solution containing 5% polyethylene glycol (PEG)-6000, and a 50 mL solution was used for each plant. After treatment, the leaf samples were collected at 0 h, 6 h, 12 h, 24 h, and 48 h, respectively. With heat and cold treatments, the plants at the four-week-old stage were exposed to 42 °C and 4 °C, respectively. The leaf samples were then collected at various time points after treatments, including 0 h, 1 h, 3 h, 6 h, 9 h, 12 h, and 24 h. Three biological replicates were performed for each treatment.

### 4.2. Identification of HMGB Genes in Tomato

To identify the HMGB members in tomato, all tomato protein sequences were downloaded from the Solanaceae genome, and the BLAST searches were performed by TBtools software (Version 2.096) using the full-length amino acid sequences of *Arabidopsis* and rice HMGBs with a blast E-value of 10^−5^ [37]. Then, the retrieved HMGB candidate sequences were submitted to the InterPro website (https://www.ebi.ac.uk/interpro/ accessed on 21 May 2023) to annotate the structural domains [38], and only the candidates containing the HMG-boxes were referred to as HMGB homologs.

TBtools software (Version 2.096) was used to analyze the physicochemical properties of tomato SlHMGB proteins, including molecular weight and isoelectric point [39]. Subcellular localization prediction was performed using the WoLF PSORT (https://www.genscript.com/wolf-psort.html, accessed on 11 November 2022) [40].

### 4.3. Gene Structure, Chromosome Mapping, and Syntenic Analysis

Tomato genome annotation files were used to obtain CDS, exon, and UTR information, which were then submitted to the Gene Structure Display Server online software (https://gsds.gao-lab.org/, accessed on 8 February 2023) to map the gene structure schematically [41].

The chromosome location of the *SlHMGB* genes was detected by TBtools software using the tomato genome annotation file. Subsequently, a map was generated through Mapchart software (Version 2.32) to visualize the distribution of *SlHMGB* genes on the chromosomes [42].

Genome sequence files and annotation files of different species (*Solanum lycopersicum*, *Solanum pennellii*, *Solanum tuberosum*, *Nicotiana tabacum*, *Capsicum annuum*, *Arabidopsis thaliana,* and *Oryza sativa*) were downloaded from SGN (https://solgenomics.net/, accessed on 20 June 2023), TAIR (https://www.arabidopsis.org/, accessed on 20 June 2023), and MSU-RGAP (http://rice.uga.edu/, accessed on 20 June 2023) databases. Then the syntenic analysis was performed using the One Step MCScanX-Super fast program within the TBtools software (Version 2.096) package, and the syntenic plots between tomato and other species were visualized by the Dual Synteny Plot program with in the TBtools (Version 2.096). The co-linear *SlHMGB* gene pairs were highlighted [43].

### 4.4. Protein Structural Domain and Motif Analysis

The sites of protein structural domains were obtained according to the NCBI online software (https://www.ncbi.nlm.nih.gov/Structure/cdd/wrpsb.cgi, accessed on 11 March 2023) [44], and then the DOG software (Version 2.0) was used to draw the schematic diagram of protein structural domains [45].

A comparison of major structural domains of tomato HMGB protein, including HMG-box, ARID, and SSRP elements, was performed using the MEME online program (https://meme-suite.org/meme/tools/meme, v5.1.1, accessed on 11 March 2023) [46].

### 4.5. Phylogenetic Analysis

To further understand the evolutionary relationship of SlHMGB with HMGB in other species, HMGB protein sequences of *Arabidopsis thaliana* (At), *Oryza sativa* (Os), *Solanum melongena* (Sm), *Capsicum annuum* (Ca), *Solanum Pennellii* (Sp), *Nicotiana tabacum* (Nt), and *Solanum tuberosum* (St) were obtained from the Solanaceae Genomics Network and NCBI databases. The phylogenetic tree was constructed using the neighbor-joining method in MEGA-X software (Version 10.2.6) [47]. The bootstrap method was selected for the test of phylogeny, the number of repetitions was set to 1000, and the Poisson model was selected for the Model/Method [48]. Subsequently, the phylogenetic tree was annotated in detail by the iTOL tool (https://itol.embl.de/, accessed on 15 April 2023) [49].

### 4.6. Cis-Acting Elements Prediction

The 2000 bp sequence upstream of the promoter of the *SlHMGB* gene was extracted by TBtools and then submitted to the online database PlantCARE (https://bioinformatics.psb.ugent.be/webtools/plantcare/html/, accessed on 20 April 2023)to predict the *cis*-acting elements [50].

### 4.7. Expression Analyses by Quantitative Real-Time PCR (qRT-PCR)

Total RNA from different samples was extracted by the Trizol method and then reverse transcribed into cDNA using the TransGen EasyScript One-Step cDNA Synthesis SuperMix kit (TransGen Biotech, Beijing, China). Primers were designed for the target genes by DNAMAN software (Version 6.0.3.99)and then amplified by fluorescence qRT-PCR using the *PerfectStar* Green qPCR SuperMix kit with cDNA as a template on an ABI7500 machine. Finally, the relative expression of target genes was calculated by 2^−ΔΔCt^ relative quantitative analysis, and the experiment was performed with three biological replicates. The primers are listed in Appendix A [40].

### 4.8. Subcellular Localization

Four *SlHMGB* genes were cloned and fused to the pPZP211-GFP vector. Then, the constructs were introduced into *Agrobacterium* (GV3101). Then, the recombinant Agrobacterium cells were injected into tobacco (*Nicotiana benthamiana*) leaves. After transformation, the tobacco plants were incubated at 25 °C in darkness for 48 h. The fluorescence signals were observed using an LSM880 laser scanning confocal fluorescence microscope (Carl Zeiss, Oberkochen, Germany) [51].

### 4.9. Statistical Analysis

Data were analyzed by Microsoft Excel 2021 using three biological replicates, and error bars represent the standard deviation (SD). The significance of differences was assayed using a two-tailed Student’s *t*-test (* *p* < 0.05; ** *p* < 0.01) with SPSS 26.0.

## 5. Conclusions

In this study, we identified a total of eleven *SlHMGB* genes, which can be divided into four categories based on the phylogenetic analysis. All SlHMGB proteins contain the conserved HMG-box domain, a characteristic structure of the HMGB family. Syntenic analysis suggested that the HMGB homologs between tomato and other dicots share close evolutionary relationships. Analyses of *cis*-elements and gene expression patterns imply that *SlHMGB* genes may play a role in regulating organ growth, development, and response to multiple abiotic stresses in tomato. Furthermore, subcellular localization detected nuclear localization of SlHMGB2, SlHMGB3, or SlHMGB4. Thus, our data provide valuable insights into the potential biological functions of *SlHMGB* genes in tomato.

## Figures and Tables

**Figure 1 ijms-25-05850-f001:**
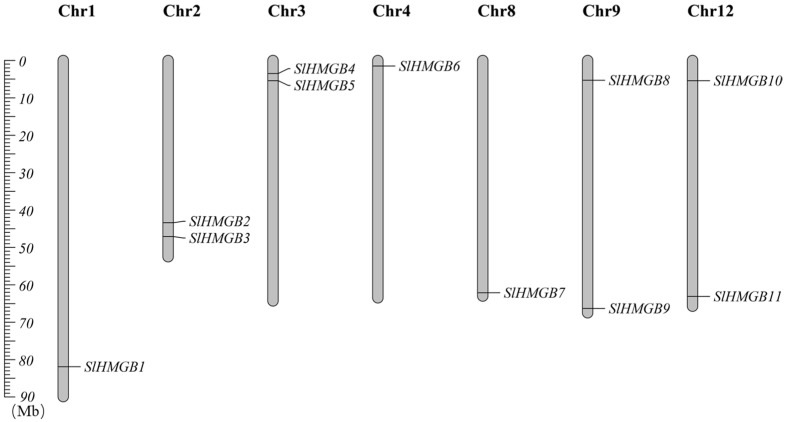
Chromosomal localization of *SlHMGB* genes in tomato. The different chromosomes are indicated by vertical gray bars with different lengths. The positions of the genes are shown in order from top to bottom.

**Figure 2 ijms-25-05850-f002:**
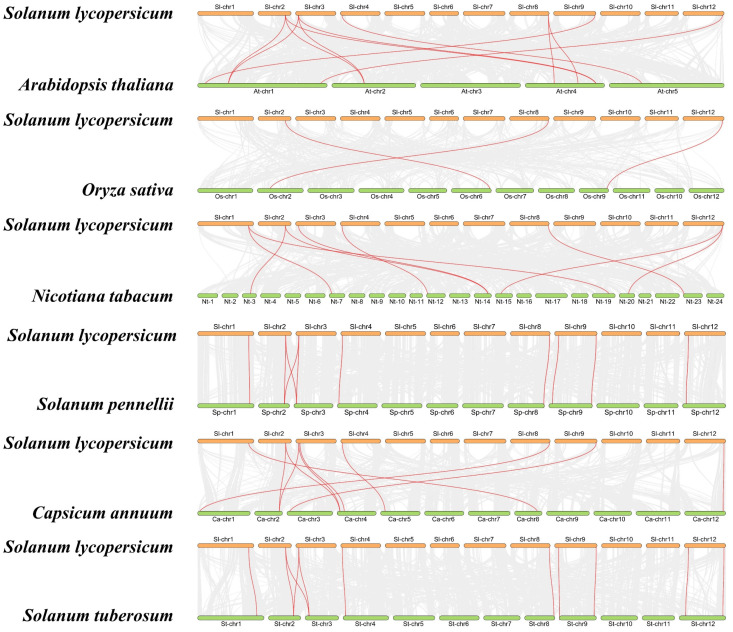
Syntenic analyses of *HMGB* genes between tomato and other plant species. Gray lines in the background indicate all syntenic blocks between two genomes. The syntenic *HMGB* gene pairs are marked by red lines.

**Figure 3 ijms-25-05850-f003:**
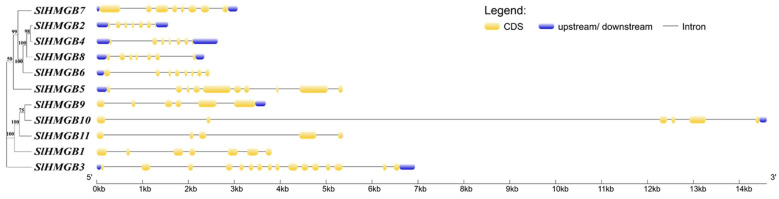
Structure of *SlHMGB* genes. The coding sequence (CDS), upstream/downstream untranslated region (UTR), and intron are indicated by yellow boxes, blue boxes, and black lines, respectively.

**Figure 4 ijms-25-05850-f004:**
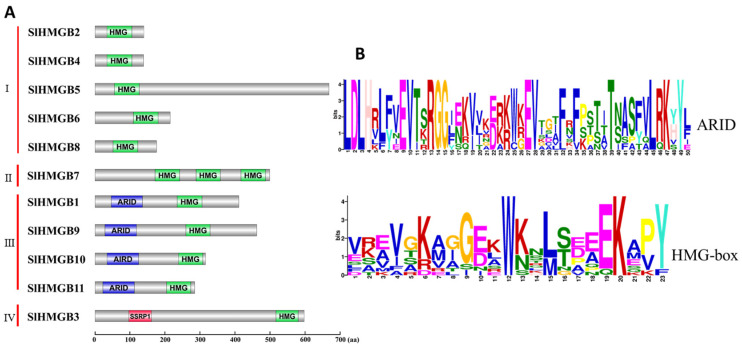
Analysis of conserved domains of SlHMGB proteins. (**A**) Protein domain structures of SlHMGB proteins. Based on the different domain structures, the SlHMGB family was divided into four groups. HMG-box, ARID, and SSRP1 domains are marked in different colors. (**B**) Amino acid sequences of ARID and HMG-box domains of SlHMGB proteins. The logos of the ARID domain were generated from the alignment of four SlHMGB sequences in group III by the MEME website, and the HMG-box logos were obtained from all 11 SlHMGB members. The font size represents the conservation level of amino acids. The larger the font size, the higher the conservation.

**Figure 5 ijms-25-05850-f005:**
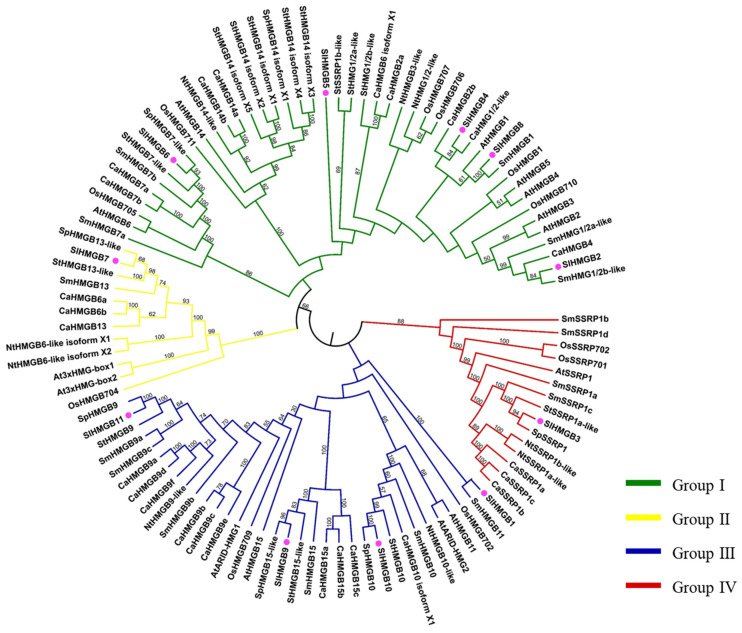
Phylogenetic analysis of HMGB homologs in different species. The clades of groups I, II, III, and IV are marked in green, yellow, blue, and red, respectively. Tomato SlHMGBs are indicated by pink balls. At, *Arabidopsis thaliana*; Ca, *Capsicum annuum*; Nt, *Nicotiana tabacum*; Os, *Oryza sativa*; Sl, *Solanum lycopersicum*; Sp, *Solanum pennellii*; St, *Solanum tuberosum*.

**Figure 6 ijms-25-05850-f006:**
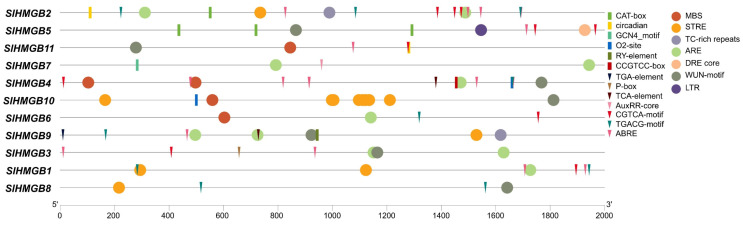
*Cis*-element analysis in the promoters of *SlHMGB* genes. The elements belonging to plant growth and development, phytohormone responsiveness, and stress responsiveness categories are represented by rectangles, triangles, and circles with different colors, respectively.

**Figure 7 ijms-25-05850-f007:**
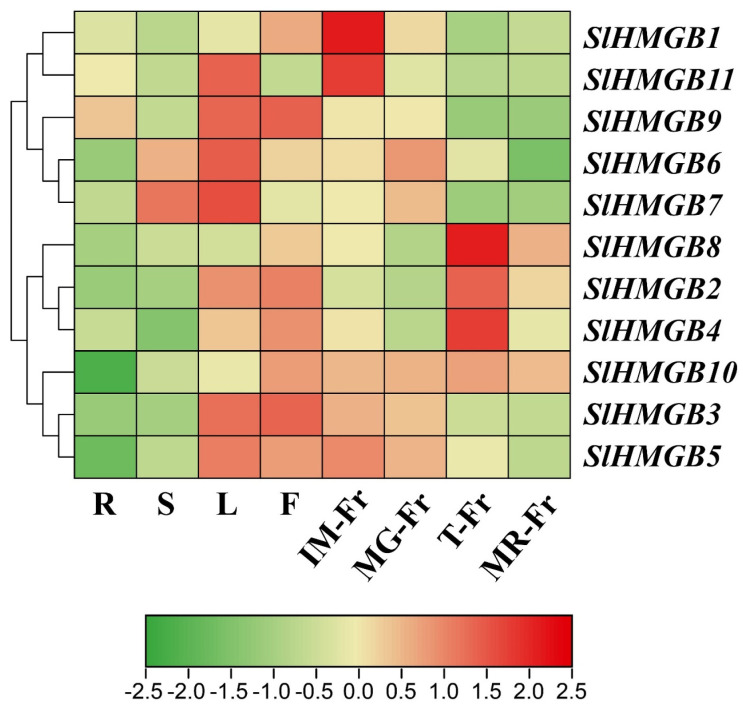
The tissue-specific expression patterns of *SlHMGB* genes. Log_2_ normalization of qRT-PCR data was performed by TBtools software (Version 2.096) and used to construct the heatmap. R, roots; S, stems; L, leaves; F, flowers; IM-Fr, immature fruits; MG-Fr, mature green fruits; T-Fr, fruits at the turning stage; MR-Fr, mature red fruits. The numerical values in the box represent the values of log_2_ normalization of qRT-PCR data. The larger the numerical value, the higher the gene expression level.

**Figure 8 ijms-25-05850-f008:**
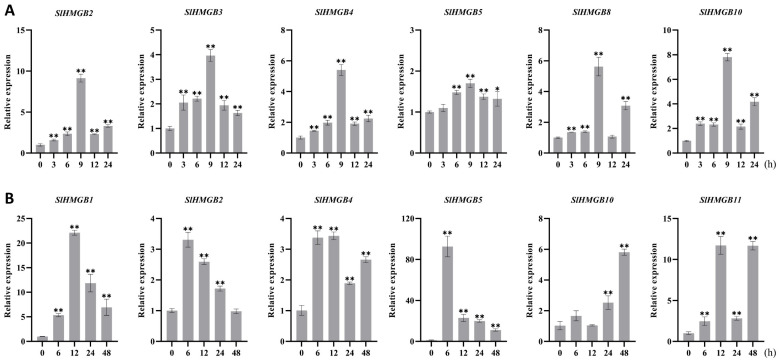
qRT-PCR analyses of *SlHMGB* genes under salt and drought stresses. (**A**,**B**) Relative expression levels of *SlHMGBs* under salt (**A**) and drought (**B**) treatments. Each value is the mean ± SD of three biological replicates. The asterisks indicate significant differences between abiotic stress treatments (different time points) and the control (0 h) by Student’s *t*-tests (* *p* < 0.05, ** *p* < 0.01).

**Figure 9 ijms-25-05850-f009:**
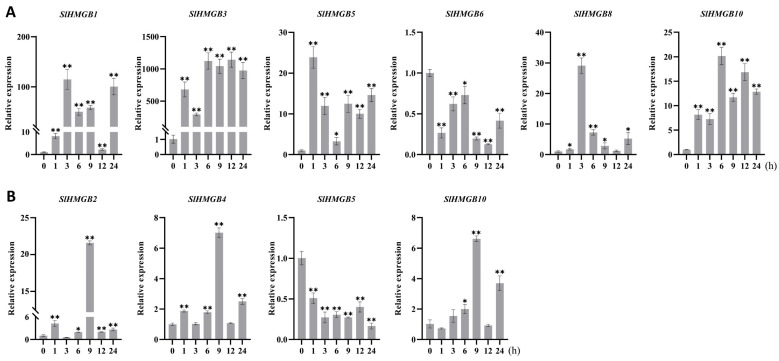
qRT-PCR analyses of *SlHMGBs* under heat and cold stresses. (**A**,**B**) Relative transcript levels of *SlHMGBs* under heat (**A**) and cold (**B**) treatments. Values are the means ± SD of three biological replicates. The asterisks indicate significant differences between abiotic stress treatments (different time points) and the control (0 h) by Student’s *t*-tests (* *p* < 0.05, ** *p* < 0.01).

**Figure 10 ijms-25-05850-f010:**
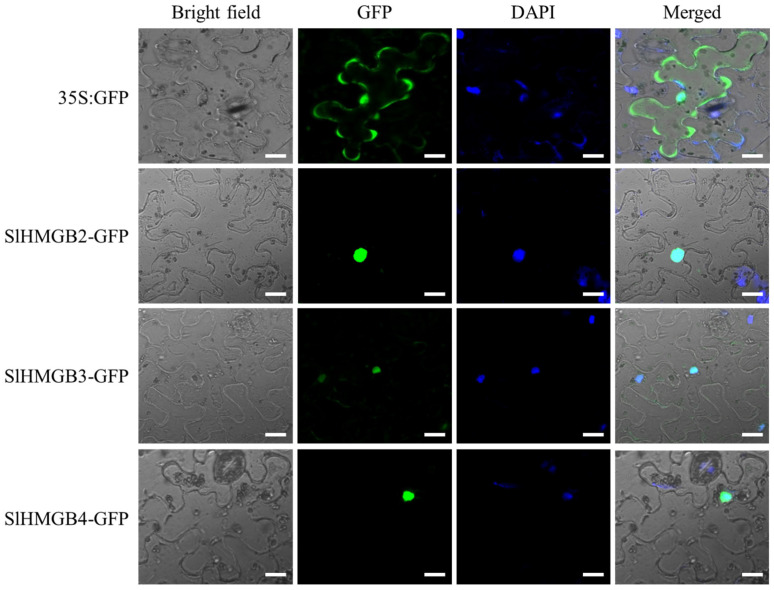
Subcellular localization of SlHMGB proteins. The constructs of *35S:GFP*, *SlHMGB2-GFP*, *SlHMGB3-GFP*, *SlHMGB4-GFP*, and DAPI were co-transformed into the tobacco leaves. 35S:GFP served as a control. The fluorescence signals were detected by a laser-scanning confocal microscope. Bars = 20 μm.

**Table 1 ijms-25-05850-t001:** Profiles of the *SlHMGB* gene family in tomato.

Gene Name	Gene ID	Genomic Sequence (bp)	CDS Sequence (bp)	Protein			Subcellular Localization Prediction
				Length (aa)	MW (kDa)	pI	
*SlHMGB1*	Solyc01g100620	3819	1236	411	45.93	8.77	Nuclear
*SlHMGB2*	Solyc02g082700	1556	423	140	15.66	7.76	Nuclear
*SlHMGB3*	Solyc02g087710	6929	1920	639	71.09	5.56	Nuclear
*SlHMGB4*	Solyc03g032130	2637	420	139	15.52	6.21	Nuclear
*SlHMGB5*	Solyc03g043600	5360	2007	668	74.46	7.46	Nuclear
*SlHMGB6*	Solyc04g008820	2457	648	215	23.74	5.46	Nuclear
*SlHMGB7*	Solyc08g082070	3069	1500	499	58.69	9.27	Nuclear
*SlHMGB8*	Solyc09g014620	2346	531	176	19.96	5.62	Nuclear
*SlHMGB9*	Solyc09g091960	3680	1389	462	52.89	6.22	Nuclear
*SlHMGB10*	Solyc12g016190	14,596	951	316	35.84	9.3	Nuclear
*SlHMGB11*	Solyc12g094440	5362	858	285	32.89	9.44	Nuclear

MW, molecular weight; pI, isoelectric point.

## Data Availability

The data supporting the findings of this study are available within the article.

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
