# Peer review of "Genome-Wide Identification and Expression Analysis of the High-Mobility Group B (HMGB) Gene Family in Plant Response to Abiotic Stress in Tomato"

_ijms, 2024, doi:10.3390/ijms25115850_

Round 1
Reviewer 1 Report
Comments and Suggestions for Authors
In the manuscript entitled “Genome-wide identification and expression analysis of high mobility group B (HMGB) gene family in plant response to abiotic stress in tomato”, eleven tomato high mobility group B (HMGB) genes were identified. The authors analyzed the gene structure, domain conservation, cis-acting elements, and collinearity with other species of the tomato SlHMGB gene family. Additionally, using qRT-PCR, the authors extensively investigated the expression patterns of SlHMGB under drought, salt, high temperature, and low temperature stress conditions, as well as tissue-specific expression patterns. However, there are still a lot of queries that need to be addressed.
1. Line 113 The minimum length of CDS is 420 bp
2. Line 114 The minimum length of protein is 139 aa
3. Table1 What is the unit of protein sequence length
4. Figure1 What is the unit of the scale
5. Figure4 Please add the x-axis to represent length. Additionally, the font size represents the conservation of amino acids rather than the conservation of domains
6. Figure 7 Please add the meaning of the numerical values in the figure
7. Figure 10 Please check 'fig.' or 'figure'. Change 35S-GFP to 35S:GFP. Change SlHMGB2:GFP to SlHMGB2-GFP. The GFP and DAPI of SlHMGB3 and SlHMGB4 appear to be exchanged.
8. Line 411 Please add the URLs for the databases of each species
Comments on the Quality of English LanguageEnglish writing need to be improved by native speaker
Author Response
Dear reviewer,
We upload the response as a Word file, please see the attachment.
Thanks for your careful reviews and constructive suggestions concerning our manuscript.
Sincerely,
Jinhui Zheng and Yan Zhang

Reviewer 2 Report
Comments and Suggestions for Authors The genome-wide investigation of HMGB genes in tomatoes is a critical study; nonetheless, several comments must be addressed.1- The abstract, particularly the results section, must address the gene expression of the genes under research.
2- Bustin et al. (2001), line 33, this reference must be the same of the numerical references.
3-In line 40 (in mammals), you mentioned the gene family of your study. What about this family of plants?
4-Line 373: The tomato cultivar Micro-Tom was used in this study. The plants were cultivated in a greenhouse; I did not notice your experimental settings, such as day/night hours and others.
5- What E-value have you used to make your blast?
6- Please rephrase line 15 of the abstract, mentioning the methods you use in your experiment.
7-Carefully double-check the manuscript's English. Comments on the Quality of English Language Carefully double-check the manuscript's English.
Author Response

(The authors gave the same response as above.)

Reviewer 3 Report
Comments and Suggestions for Authors
Genome-wide identification and expression analysis of high mobility group B (HMGB) gene family in plant response to abiotic stress in tomato
Jinhui Zheng, Huimeng Tang, Jianquan Wang, Yue Liu, Lianjing Ge, Guobiao Liu, Qinghua Shi, Yan Zhang
Authors provide new information about an important, so far little-researched gene family in tomatoes, which is why the study is definitely timely and novel.
A few comments and minor suggestions:
1.In the introduction, the section on mammals and human aspects is irrelevant to the study, so I recommend leaving it out.
2. The test methods fully meet the requirements of modern genetic analyses. The authors analyzed the members of the gene family in many ways. They identified individual members of the gene family in tomato, then mapped the gene structure and outlined their chromosomal position. During their phylogenetic analysis, they established the relationships between similar sequences in other species and the sequences found in tomatoes.
3.They not only performed tissue- or organ-specific analyses, but also covered plants subjected to various abiotic stresses.
This analysis is particularly valuable, as it proves that these genes are expressed under various abiotic stresses.
4. The only shortcoming of this all-around excellent study is that it does not attempt to find an explanation for the gene expression results obtained. Although they cite references found in other plant species, do not justify the results obtained. In the same way, they didn’t do this with the gene expression results obtained under stress.
It would be proposed to justify the obtained results with explanations and assumptions, and where literature data is also available, to carry out a discussion with them. I mean that they should not only write that, e.g. under a given stress, how many genes were expressed at a higher level, even please try to rationally justify the results achieved with plant physiological explanations, or even with changes in biochemical or signaling pathways.
In addition to these corrections, I recommend the publication of the study.
Author Response

(The authors gave the same response as above.)
